



# Temporal Evolution of the Bromine Alpha Factor and Equivalent Effective Stratospheric Chlorine in Future Climate Scenarios

J. Eric Klobas[1], Debra K. Weisenstein[1], Ross J. Salawitch[2], and David M. Wilmouth[1]

[1]Harvard John A. Paulson School of Engineering and Applied Sciences, Harvard University, Cambridge MA, USA
[2]Department of Atmospheric and Oceanic Science, Department of Chemistry and Biochemistry, and Earth System Science Interdisciplinary Center, University of Maryland, College Park, MD, USA

**Correspondence:** J. Eric Klobas (klobas@huarp.harvard.edu)

**Abstract.** Future trajectories of the stratospheric trace gas background will alter the rates of bromine- and chlorine-mediated catalytic ozone destruction via changes in the partitioning of inorganic halogen reservoirs and the underlying temperature structure of the stratosphere. The current formulation of the bromine alpha factor, the ozone-destroying power of stratospheric bromine atoms relative to stratospheric chlorine atoms, is invariant with climate state. Here, we refactor the bromine alpha

factor, introducing climate normalization to a benchmark climate state, and reformulate Equivalent Effective Stratospheric Chlorine (EESC) to reflect changes in the rates of both chlorine- and bromine-mediated ozone loss catalysis with time. We show that the ozone-processing power of the extrapolar stratosphere is significantly perturbed by future climate assumptions. Furthermore, we show that our EESC-based estimate of the extrapolar ozone-recovery date is in closer agreement with extrapolar ozone recovery dates predicted using more sophisticated 3-D chemistry-climate models than prior formulations of EESC

that employ climate-invariant values of the bromine alpha factor.

## 1  Introduction

Anthropogenic emissions of ozone-destroying halocarbons have declined significantly since the implementation of the Montreal Protocol on Substances that Deplete the Ozone Layer (and its subsequent amendments); however, the stratospheric inorganic halogen background still remains elevated relative to levels prior to the first observations of the seasonal Antarctic ozone

hole due to the exceptionally long lifetimes of the inorganic halogen precursor compounds (World Meteorological Organization, 2018). Recovery of the halogen content of the stratosphere to the levels representative of the benchmark year 1980 is estimated to occur some time around the year 2060 in the extrapolar regions (Newman et al., 2006; Engel et al., 2018; World Meteorological Organization, 2018); however, 3-D CCM simulations predict ozone recovery dates up to two decades sooner (Dhomse et al., 2018) as halogen inventory recovery is an imperfect proxy for ozone recovery.

The vast majority of inorganic chlorine in the stratosphere is present in the reservoir forms HCl and ClONO$_2$. In the lower and middle stratosphere, it is typical that only a few percent of inorganic chlorine is present in active forms, such as ClO. Active halogen radical species participate in ozone-destroying chemical reaction cycles, such as the catalytic cycle presented





in reactions R1 –R3 below, in which inorganic chlorine is rapidly interconverted between the chlorine radical and the chlorine monoxide radical.

$$Cl + O_3 \rightarrow ClO + O_2 \tag{R1}$$

$$ClO + O \rightarrow Cl + O_2 \tag{R2}$$

$$Net: O_3 + O \rightarrow 2O_2 \tag{R3}$$

The gas-phase conversion of the dominant inorganic chlorine reservoirs to their active, ozone-destroying forms (reaction R4) is too slow to be of atmospheric importance; however, heterogeneous reactions on the surfaces of stratospheric aerosols (Solomon et al., 1986; Brasseur et al., 1990), as indicated in reactions R5–R7, can be sufficiently fast to enable significant engagement of $ClO_x$ ozone-depletion cycling.

$$HCl + ClONO_2 \xrightarrow{homogeneous} HNO_3 + Cl_2 \tag{R4}$$


$$HCl + ClONO_2 \xrightarrow{heterogeneous} HNO_3 + Cl_2 \tag{R5}$$

$$ClONO_2 + H_2O \xrightarrow{heterogeneous} HNO_3 + HOCl \tag{R6}$$

$$HOCl + HCl \xrightarrow{heterogeneous} H_2O + Cl_2 \tag{R7}$$

Mechanisms of $BrO_x$-mediated ozone depletion are much less dependent on the surrounding environment than mechanisms mediated by $ClO_x$. This is because inorganic reservoirs of bromine are significantly less stable, enhancing the quantity of reactive halogen available for ozone processing. Bromine is up to two orders of magnitude more likely to be found in its active form than chlorine, depending on the physicochemical environment (Wofsy et al., 1975; Salawitch et al., 2005). Additionally,

unlike the chlorine cycle presented in reactions R1–R3 which requires the presence of atomic oxygen and is accordingly slow in the lower stratosphere or in regions of low actinic flux, catalytic processing of ozone facilitated by the addition of bromine is effective in these regions. Reactions R8–R11, the coupled odd bromine-chlorine cycle, and reactions R12–R16, the coupled odd bromine-hydrogen cycle, are examples of this chemistry in which atomic oxygen is not involved. The bromine interfamily





cycles are responsible for a similarly-sized fraction of global lower stratospheric ozone loss as the chlorine cycle (reactions R1 – R3) (Salawitch et al., 2005; World Meteorological Organization, 2018; Koenig et al., 2020). This large fractional share of ozone destruction chemistry occurs despite the fact that bromine is approximately two orders of magnitude less abundant than chlorine as a consequence of (a) the larger fraction of reactive bromine available at a given mixing ratio and (b) the catalytic reaction channels made accessible by the weaker bromine-oxygen molecular bond (Yung et al., 1980; McElroy et al., 1986; Brune and Anderson, 1986; World Meteorological Organization, 2018).

$$ClO + BrO \rightarrow products \rightarrow Cl + Br + O_2 \tag{R8}$$

$$Cl + O_3 \rightarrow ClO + O_2 \tag{R9}$$

$$Br + O_3 \rightarrow BrO + O_2 \tag{R10}$$

$$Net: 2O_3 \rightarrow 3O_2 \tag{R11}$$

$$HO_2 + BrO \rightarrow HOBr + O_2 \tag{R12}$$

$$HOBr + h\nu \rightarrow Br + OH \tag{R13}$$

$$Br + O_3 \rightarrow BrO + O_2 \tag{R14}$$

$$OH + O_3 \rightarrow HO_2 + O_2 \tag{R15}$$

$$Net: 2O_3 \rightarrow 3O_2 \tag{R16}$$

The bromine alpha factor, $\alpha_{Br}$, is a metric that quantifies the ozone-depleting efficiency of a bromine atom relative to chlorine. This quantity is defined either as the ratio of ozone loss processing rates, as in Eq. (1) or as the ratio of the overall change in





ozone abundance on a per-halogen-atom basis per Eq (2). In both formulations, $\alpha_{\mathrm{Br}}$ is computed as a function of calendar

date, $t$, and location in the atmosphere, $\rho$. Daniel et al. (1999) demonstrate that both equations provide identical results when

changes in ozone are dominated by chemical rather than dynamical processes.

$$\alpha_{\mathrm{Br}}(t,\rho) = \frac{\frac{\Delta \mathrm{O}_3(t,\rho)}{\Delta t}\,(\mathrm{Br\ rxns})\,/\,\Delta \mathrm{Br}(t,\rho)}{\frac{\Delta \mathrm{O}_3(t,\rho)}{\Delta t}\,(\mathrm{Cl\ rxns})\,/\,\Delta \mathrm{Cl}(t,\rho)} \tag{1}$$

$$\alpha_{\mathrm{Br}}(t,\rho) = \frac{\Delta \mathrm{O}_3(t,\rho)/\Delta \mathrm{Br}(t,\rho)}{\Delta \mathrm{O}_3(t,\rho)/\Delta \mathrm{Cl}(t,\rho)} \tag{2}$$

Values of $\alpha_{\mathrm{Br}}$ vary strongly as a function of pressure, latitude, and season. This variance is primarily a function of (a)

chemical environment, (b) prevailing actinic flux, (c) aerosol surface area, and (d) temperature (Solomon et al., 1992; Danilin

et al., 1996; Ko et al., 1998; Daniel et al., 1999). Frequently, $\alpha_{\mathrm{Br}}$ is reported as an effective value for the stratospheric column,

computed in a similar manner as in Eq. (1) or Eq. (2), the key difference being that $\rho$ represents the position of the stratospheric

column. Likewise, it is common to provide a regional-annual average column $\alpha_{\mathrm{Br}}$, which is computed as the average of column

$\alpha_{\mathrm{Br}}$ values for all locations within a specified region across a calendar year. Global-annual average column values for $\alpha_{\mathrm{Br}}$ are

currently estimated between $60 - 65$, depending on the model employed and the chemoclimatic boundary conditions (World

Meteorological Organization, 2018; Sinnhuber et al., 2009). Values of $\alpha_{\mathrm{Br}}$ tend toward a minimum at the equator, maximizing

in the boreal summer. Denitrification and heterogeneous activation produce a minimum in $\alpha_{\mathrm{Br}}$ during the austral springtime.

In vertical profiles, $\alpha_{\mathrm{Br}}$ tends to maximize in the lower stratosphere where reactive chlorine is less prevalent than in the middle

stratosphere.

     The quantity $\alpha_{\mathrm{Br}}$ is especially useful for the determination of parameterized estimates of the budget of reactive inorganic

halogens given a mixture of halogen-containing halocarbons of an arbitrary mean age, as in the metric of Equivalent Effective

Stratospheric Chlorine (EESC). This quantity expresses the ozone-depleting power of a parcel of well-mixed stratospheric trace

gases as a function of mean stratospheric age of the parcel, $\Gamma$, and the trace gas background of the stratosphere at time $t$ (Daniel

et al., 1995; Newman et al., 2007). Equation 3 provides the most recently suggested formulation of EESC, in which $\overline{f_i}(\Gamma)$ is

the time-independent fractional release factor for species $i$ for a parcel of air with mean age $\Gamma$, which contains $n_{i,Cl}$ chlorine

atoms and $n_{i,Br}$ bromine atoms, scaled by $\alpha_{\mathrm{Br}}(t,\Gamma)$, where it is assumed that $\Gamma$ can serve as a proxy for $\rho$ (Ostermöller et al.,

2017; Engel et al., 2018). Inside the integral, the mixing ratio of species $i$ is computed for each element comprising the age

spectrum and normalized to the contribution of that element to the age spectrum. The tropospheric mixing ratio of species $i$,

$\chi_{0,i}$ is adjusted to account for transit time within the stratosphere, $t'$, and multiplied by the normalized release-weighted transit

time distribution, $G_{N,i}^{\#}(\Gamma^{\#},t')$, where $\Gamma_i^{\#}$ is the mean age of halogen-atom release.

$$\mathrm{EESC}(t,\Gamma) = \sum_i \overline{f_i}(\Gamma)\,[n_{i,\mathrm{Cl}} + \alpha_{\mathrm{Br}}(t,\Gamma)\cdot n_{i,\mathrm{Br}}] \int_0^\infty \chi_{0,i}(t-t')G_{N,i}^{\#}(t',\Gamma_i^{\#})dt' \tag{3}$$

EESC is frequently employed to approximate the date of stratospheric ozone recovery, often by using graph theory to determine

when stratospheric chlorine levels will return to the levels observed in 1980 as a benchmark (Newman et al., 2006; World



Meteorological Organization, 2018). The technique is fast and simple: EESC is calculated as a function of location in the stratosphere (for which $\Gamma$ is a proxy) and future date, following which a horizontal line is propagated in time at the value of EESC in 1980, and the intercept of the two traces is interpreted as the date of halogen recovery (and, it follows, the probable date of ozone recovery). The extrapolation is built on the assumptions that, as the climate evolves: (1) the alpha factor remains constant and (2) the amount of ozone destroyed by chlorine, on a per-chlorine-atom basis, also remains constant. However,

projections of the future physicochemical state of the stratosphere do not necessarily provide for these two assumptions to be true. Indeed, the envelope of future projections (e.g., RCP and SSP scenarios) of emissions of $CH_4$, $N_2O$, $CO_2$, among other relevant species, indicate that it is nearly certain that these two assumptions will not be true, especially in the extrapolar stratosphere.

Significant variations between different climate models and possible states of the future atmosphere limit the skill level of

model simulations in predicting ozone recovery dates (Charlton-Perez et al., 2010). These large uncertainties notwithstanding, it is understood that there may be a super-recovery of global stratospheric ozone in the future as EESC declines and the stratosphere cools (Austin and Wilson, 2006; Li et al., 2009; Eyring et al., 2013; Banerjee et al., 2016; Chiodo et al., 2018). The extent of super-recovery is primarily dependent on the degree by which rates of bimolecular ozone-loss processes are slowed and the rate of the termolecular formation of ozone is increased as a result of (a) local radiative cooling due to the

enhancement of the stratospheric burden of anthropogenic greenhouse gases and (b) chemical suppression of ozone loss cycling due to reactive anthropogenic greenhouse gas emission (Rosenfield et al., 2002; Waugh et al., 2009; Oman et al., 2010; Eyring et al., 2013). Future projections of ozone are also dependent on dynamical factors, such as the model response of the Brewer-Dobson circulation to greenhouse gas perturbation, which alters both the stratospheric lifetime of long-lived inorganic halogen precursors and the transport of ozone from the tropics where it is produced (Butchart et al., 2006; Plummer et al., 2010; Zubov

et al., 2013).

Dhomse et al. (2018) provide constraints on the dates stratospheric ozone might recover to year 1980 benchmark thickness using a comprehensive multi-model framework (20 models, 155 simulations) spanning multiple greenhouse gas emissions scenarios, finding that while the date of Antarctic springtime recovery is most sensitive to $Cl_y$ inventories, extrapolar column recovery dates (and to a lesser extent, the Arctic springtime recovery date) are highly sensitive to the greenhouse gas forcing

applied. In their analysis, Dhomse et al. (2018) indicate that mid-latitude ozone recovery will occur sooner in both hemispheres for scenarios with greater radiative forcing. When greenhouse gases are fixed, the dates projected for midlatitude recovery (~2060) are in close agreement with the EESC-based estimates provided in Engel et al. (2018) of 2059; however, greenhouse gas perturbations hasten projected midlatitude recovery dates in 3-D models by ~10 years in the northern hemisphere and ~20 years in the southern hemisphere (Eyring et al., 2010, 2013; Dhomse et al., 2018).

Regardless, it is known that the decay of EESC is the strongest driver of ozone recovery (Meul et al., 2014; Banerjee et al., 2016). While EESC is expected to decrease in the future, it is increasingly evident that the inorganic halogen background might be significantly perturbed by natural geological processes under certain circumstances (Klobas et al., 2017). Volcanic eruptions are now known to frequently inject small quantities of inorganic chlorine into the lower stratosphere (Carn et al., 2016), and there exists evidence for the periodic and profound volcanic injection of inorganic chlorine in the ice core record





**Table 1.** Experiment Schedule[a]

| experiment prefix | decades[b] | climatology | $CFCl_3A$ (pptv) | $CFBr_3$ (pptv) |
|:---:|:---:|:---:|:---:|:---:|
| bkg | [1980–2010] | historical[c] | 0 | 0 |
| bkg | [2020–2100] | RCP[2.6,4.5,6.0,8.5][d] | 0 | 0 |
| Cl | [1980–2010] | historical[c] | 260 | 0 |
| Cl | [2020–2100] | RCP[2.6,4.5,6.0,8.5][d] | 260 | 0 |
| Br | [1980–2010] | historical[c] | 0 | 2.6 |
| Br | [2020–2100] | RCP[2.6,4.5,6.0,8.5][d] | 0 | 2.6 |

[a] All permutations of bracketed parameters were evaluated.

[b] Constant year for each decade (e.g., 1980, 1990, 2000)

[c] Informed by Fleming et al. (1999)

[d] Informed by Meinshausen et al. (2011) and Watanabe et al. (2011)

(Zdanowicz et al., 1999) following large, explosive eruptions. Additionally, it is now apparent that volcanic bromine and iodine may partition more effectively to the stratosphere than volcanic chlorine (Theys et al., 2009, 2014; Schönhardt et al., 2017; Gutmann et al., 2018). The expected enhancement in ozone-loss processing rates due to additional volcanogenic inorganic halogens following a future, large, halogen-rich explosive eruption is not well constrained, partially because the temporal evolution of the ozone processing rates of bromine relative to chlorine is largely unknown.

In this work, we present the first assessment of column $\alpha_{Br}$ in future climate change scenarios. Additionally, we evaluate the sensitivity of column $\alpha_{Br}$ to prescribed perturbations of reactive greenhouse gases while anthropogenic halocarbons slowly decay as the century progresses. We then refactor $\alpha_{Br}$, such that estimates of EESC can more accurately be related to the ozone-destroying power of the inorganic halogen background of the stratosphere given a particular benchmark date. Finally, we show that this method provides much better agreement between EESC-based estimates and 3-D CCM estimates of ozone
recovery to the 1980 benchmark date.

## 2   Model, Experiment, and Validation

The AER-2D chemical transport model was employed with 19 latitudes (90°S–90°N) and 51 levels (1000–0.2 hPa) for this work. The model includes 104 chemical species, accounting for $F_y$, $Cl_y$, $Br_y$, $I_y$, $NO_y$, $HO_x$, $O_x$, $SO_x$, and $CHO_x$ chemistry. Chemical reactions (314 kinetic reactions and 108 photochemical reactions) were computed using rate constants and cross sec-
tions as recommended in the most recent (2015) JPL data evaluation (Burkholder et al., 2015). Additionally, the model features fully-prognostic aerosol microphysics and chemistry (e.g., nucleation, coagulation, condensation/evaporation, sedimentation, and heterogeneous chemical interactions in 40 sectional size bins). Future emissions of greenhouse gases were informed by the Representative Concentration Pathway framework (Van Vuuren et al., 2011; Meinshausen et al., 2011). Future climatological boundary conditions were obtained from the corresponding RCP experiments of MIROC-CHEM-ESM, an Earth System Model





with stratospheric chemistry. Future halocarbon inventories were informed by Table 6-4 of the 2018 WMO Scientific Assessment of Ozone Depletion (World Meteorological Organization, 2018) with an additional 5 pptv stratospheric bromine from very-short lived bromocarbons (Wales et al., 2018). Experiments performed in the historical past were informed by historical climatologies obtained from Fleming et al. (1999).

    Halogen perturbation scenarios were prepared in the manner of Daniel et al. (1999). Namely, CFC-11 proxy molecules

($CFCl_3A$ and $CFBr_3$) were constructed to provide identical transport and release of halogen atoms between model runs. For bookkeeping purposes, this was done for both chlorine and bromine delivery (e.g., the molecule labeled as $CFCl_3A$ has the same chemical kinetics and photolysis rates as CFC-11, providing 3 chlorine atoms upon decomposition, but can be perturbed in the model separately from $CFCl_3$). Experiments were performed as outlined in Table 1. Experiments of a certain scenario (e.g., bkg2020RCP8.5, Cl2020RCP8.5, Br2020RCP8.5) were initialized from identical 20-year chemical-climatological spun-

up boundary conditions. Evaluations were conducted at constant chemical and climatological conditions corresponding to the last year of each decade (e.g., 1980, 1990, ..., 2100). Perturbation and control experiments were evaluated over the course of 20 model years, a duration determined to be an appropriate period for the perturbation gas to reach chemical-dynamical steady-state. Data analysis was conducted on the final 12 months of each experiment and control run. Perturbation gas surface mixing ratios were selected to produce global and local ozone depletion of less than 1% in each climate state relative to the

unperturbed condition to prevent instability in the chemical Jacobian.

    The model performance and experiment design were validated using calculations of $\alpha_{Br}$ in a chemistry-climate state representative of the year 2006. This climate condition has previously been evaluated for $\alpha_{Br}$ (Sinnhuber et al., 2009) using the JPL-2006 photochemical-kinetics recommendations (Sander et al., 2006). Validation runs for this work were informed by either JPL-2006 or JPL-2015 photochemical-kinetics packages. A comparison of the two model evaluations is presented in Figure 1

in which there is little qualitative difference in the annual variation in $\alpha_{Br}$ between JPL-2006 and JPL-2015 photochemical-kinetics packages. Implementation of JPL-2015 chemistry results in a general increase in column $\alpha_{Br}$ of $\sim 10$ relative to JPL-2006 contours in both polar and extrapolar regions. Our annually/globally averaged $\alpha_{Br}$ of 67 in the JPL-2006 instance compares favorably to the results of Sinnhuber et al. (2009), who report an annually/globally averaged $\alpha_{Br}$ of 64 in their analysis of the same chemistry-climate state using the same photochemistry and kinetics package. For the JPL-2015 evaluation, the

annually/globally averaged $\alpha_{Br}$ is 74, which is larger than previously reported values. These differences are likely the result of a combination of changes in chemical rates between JPL-2006 and JPL-2015, such as: (a) 2% increase in rate of $Cl + CH_4$ at 200 K, (b) 8% increase in formation rate of NO by $N_2O + O(^1D)$ at 200 K, (c) 4% increase in the rate of $Br + O_3$ at 200 K, (d) 121% increase in the rate of $CHBr_3 + OH$ at 200 K, and (e) 5% increase in the rate of $Cl + ClOOCl$ at 200 K.

## 3   Results and Discussion

### 3.1   Refactoring $\alpha_{Br}$: a new definition of EESC

Prior evaluations of $\alpha_{Br}$ were computed with static chemistry-climate states. Because the relative ozone-processing rate of bromine to chlorine is likely to change as time propagates within chemistry-climate scenarios, and also between chemistry-



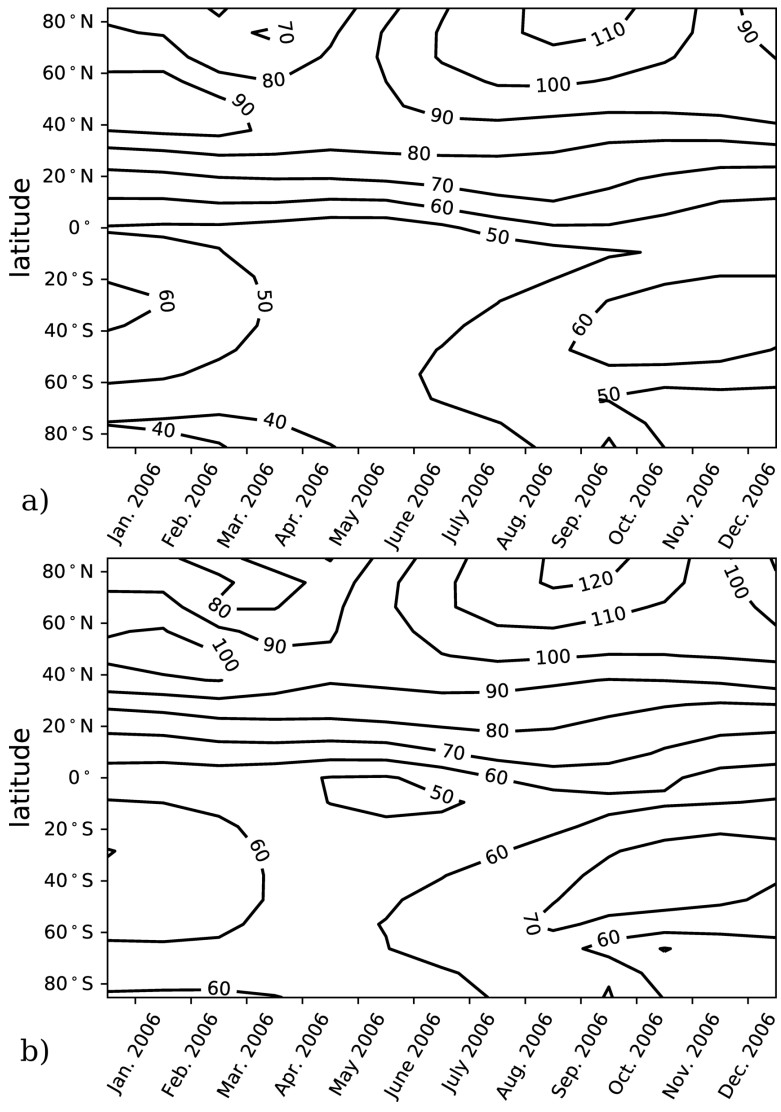

**Figure 1.** Column $\alpha_{Br}$ as a function of latitude and season for the year 2006. (a) Model results computed using JPL-2006 photochemistry and kinetics (Sander et al., 2006) for the year 2006. (b) Model results computed using JPL-2015 photochemistry and kinetics (Burkholder et al., 2015) for the year 2006.





climate scenarios at the same point in time, we add a dependence on the chemical-climate state, $\xi$, to the defintion of $\alpha_{\text{Br}}$ (Eq. (4)).

$$\alpha_{\text{Br}}(t,\rho,\xi) = \frac{\Delta O_3(t,\rho,\xi)/\Delta \text{Br}(t,\rho,\xi)}{\Delta O_3(t,\rho,\xi)/\Delta \text{Cl}(t,\rho,\xi)} \tag{4}$$

We can then replace the climate-invariant $\alpha_{\text{Br}}(t,\Gamma)$ in Eq. (3) with $\alpha_{\text{Br}}(t,\Gamma,\xi)$ to produce a climate-sensitive EESC that accounts for changes in the relative ozone-destroying efficiency of bromine to chlorine (Eq. (5)).

$$\text{EESC}(t,\Gamma,\xi) = \sum_i \overline{f_i}(\Gamma) \left[ n_{\text{i,Cl}} + \alpha_{\text{Br}}(t,\Gamma,\xi) \cdot n_{\text{i,Br}} \right] \int_0^\infty \chi_{0,i}(t-t') \cdot G_{N,i}^{\#}(t',\Gamma_i^{\#}) dt' \tag{5}$$

Furthermore, we recognize that the ozone-processing power of chlorine and bromine are independently sensitive to changes

in the physicochemical background of the stratosphere. The two variables must be separated in order to understand the evolution of the change in the processing power of bromine and chlorine as a function of climate state. To accomplish this, we define the eta factor, $\eta_{\text{Cl}}$ and $\eta_{\text{Br}}$, in Eq. (6) and Eq. (7) as the ratio of the change in ozone following the addition of chlorine or bromine at time $t$, location $\rho$, and climate state $\xi$ to the change in ozone following the same perturbation in a benchmark chemical-climate state, $\Xi$.

$$\eta_{\text{Cl}}(t,\rho,\xi,\Xi) = \frac{\Delta O_3(t,\rho,\xi)/\Delta \text{Cl}(t,\rho,\xi)}{\Delta O_3(t,\rho,\Xi)/\Delta \text{Cl}(t,\rho,\Xi)} \tag{6}$$

$$\eta_{\text{Br}}(t,\rho,\xi,\Xi) = \frac{\Delta O_3(t,\rho,\xi)/\Delta \text{Br}(t,\rho,\xi)}{\Delta O_3(t,\rho,\Xi)/\Delta \text{Cl}(t,\rho,\Xi)} \tag{7}$$

It is apparent that the definition of $\alpha_{\text{Br}}$ given in Eq. (4) can be derived from $\eta_{\text{Br}}$ and $\eta_{\text{Cl}}$ provided that the benchmark climate states are identical (Eq. (8)).

$$\alpha_{\text{Br}}(t,\rho,\xi) = \frac{\eta_{\text{Br}}(t,\rho,\xi,\Xi)}{\eta_{\text{Cl}}(t,\rho,\xi,\Xi)} \tag{8}$$

By substituting this refactored definition of $\alpha_{\text{Br}}$ into Eq. (5) for the computation of EESC, we can now quantify the ozone-depleting power of an air parcel in the stratosphere, propagated in time without bias to changes in the rates of bromine and chlorine ozone-loss catalysis (Eq. (9)) relative to the benchmark chemoclimatic state. Note again that $\rho$ has been substituted with $\Gamma$ per Engel et al. (2018).


$$\text{EESC}(t,\Gamma,\xi,\Xi) = \eta_{\text{Cl}}(t,\Gamma,\xi,\Xi) \cdot \text{EESC}(t,\Gamma,\xi) =$$

$$\sum_i \overline{f_i}(\Gamma) \left[ \eta_{\text{Cl}}(t,\Gamma,\xi,\Xi) \cdot n_{\text{i,Cl}} + \eta_{\text{Br}}(t,\Gamma,\xi,\Xi) \cdot n_{\text{i,Br}} \right] \int_0^\infty \chi_{0,i}(t-t') \cdot G_{N,i}^{\#}(t',\Gamma_i^{\#}) dt' \tag{9}$$





Equation 9 provides a more appropriate basis for a graph-theory approximation of future inorganic halogen ozone-loss processing than prior approaches because the ordinate now represents Equivalent Effective Stratospheric Chlorine normalized

to a benchmark atmospheric state rather than an instantaneous equivalent EESC with a time-varying ozone-processing power per chlorine atom.

### 3.2 Calculation of Future RCP scenario $\alpha$ and $\eta$

Packaged within the definitions of $\alpha$ and $\eta$ are both local (e.g., photochemical catalytic processing) and non-local influences on ozone abundance (e.g., dynamical effects, ozone layer self-healing effect, etc.), as illustrated in Eq. (10). These non-local

factors do not cancel out in the evaluation of $\eta_{Cl}$ per Eq. (6) or $\eta_{Br}$ per Eq. (7) as they do in the calculation of $\alpha_{Br}$ per Eq. (1) or Eq. (2), because the non-local factors at time $t$ in some evolved climate state are not likely to be the same as they were during the benchmark time period.

$$\frac{\Delta O_3(t,\rho,\xi)}{\Delta O_3(t,\rho,\Xi)} \simeq \frac{(\Delta O_3(t,\rho,\xi)_{\text{photochem.}} + \Delta O_3(t,\rho,\xi)_{\text{dyn.}})}{(\Delta O_3(t,\rho,\Xi)_{\text{photochem.}} + \Delta O_3(t,\rho,\Xi)_{\text{dyn.}})} \tag{10}$$

To avoid this complication, we employ specified dynamics corresponding to the 1978 – 2004 climatological average in order

to calculate only the photochemical component of the ozone tendency. These dynamics tend to produce less seasonal variation in $\alpha_{Br}$ in the extrapolar southern hemisphere than in the extrapolar northern hemisphere, as depicted in Figure 1. Because of the carefully controlled magnitude of the imposed ozone deficit ($\sim$1%), changes in ozone between experiment and control scenarios from all other effects can be assumed to be insignificant relative to the ozone changes produced by the chemical perturbation.

Diagnostic trajectories of the well-mixed greenhouse gases employed in the climatological perturbations are illustrated in Figure 2, constructed from data provided by Meinshausen et al. (2011). Prescribed mixing ratios of $CO_2$, which is chemically inert in this model and only perturbs ozone chemistry via thermal effects, are provided in panel (a). The trajectories of $CH_4$ in panel (b) and $N_2O$ in panel (c) are particularly noteworthy because these species are closely coupled with the ozone steady-state via changes in inorganic halogen reservoir inventories. In the instance of RCP 8.5, $CH_4$ increases nearly 2.5 times by the year

2100 from the 1980 mixing ratio, and $N_2O$ increases by a factor of 1.4 during the same time period. The reactive greenhouse gas situation in RCP 2.6 is significantly different: $CH_4$ mixing ratios decline by 19% and $N_2O$ mixing ratios increase by 14%. The intermediate scenarios, RCP 4.5 and RCP 6.0, both feature small end-of-century increases in $CH_4$ mixing ratios of less than 10%, but with modest increases during the middle-half of the 21st century. Prescribed $N_2O$ emissions increase monotonically by 24% and 35% respectively.

Values of annually-averaged extrapolar $\eta_{Cl}$ and $\eta_{Br}$ (60°S – 60°N) were computed on a decadal basis for every decade between 1990 – 2010 using historical data, and for each decade between 2020 – 2100, for each RCP scenario. For all results reported in this work, the chemistry-climatology corresponding to the year 1980 was selected as the benchmark state ($\Xi =$ 1980). These values are presented, along with the corresponding alpha factors, in Table 2 for the historical period and for future scenarios. These results are also visualized in Figure 3 for (a) extrapolar $\alpha_{Br}$, (b) extrapolar $\eta_{Cl}$, and (c) extrapolar $\eta_{Br}$. It

is immediately apparent that, while $\alpha_{Br}$ deviates by less than 10% from its 1980 value for all evaluated future atmospheres as



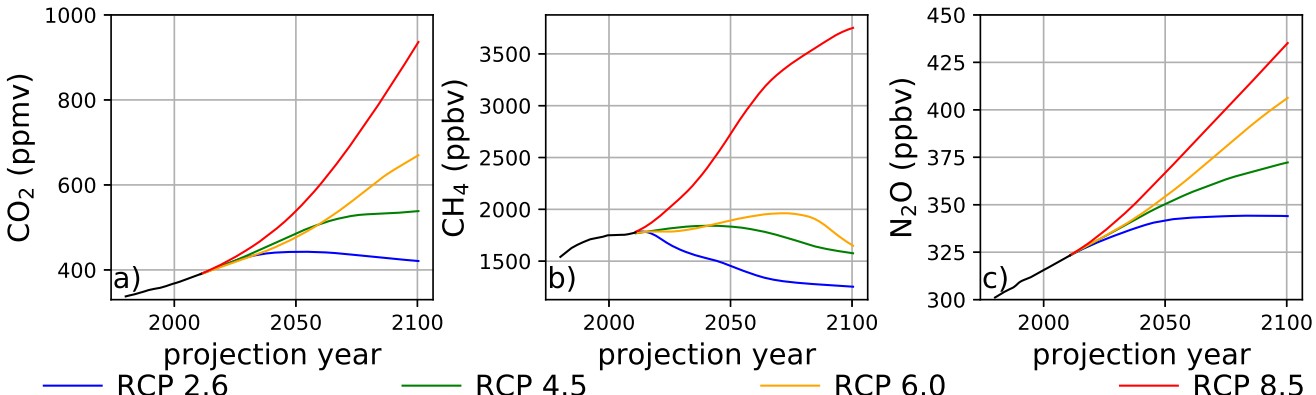

**Figure 2.** Surface mixing ratios of (a) $CO_2$, (b) $CH_4$, and (c) $N_2O$ as a function of time and RCP scenario. Data obtained from Meinshausen et al. (2011).

**Table 2.** Values of extrapolar $(60°\,S - 60°\,N)$ $\alpha_{Br}{}^a$, $\eta_{Cl}{}^b$, and $\eta_{Br}{}^b$ for historical and future scenarios

| Year | Historical | | | Year | RCP 2.6 | | | RCP 4.5 | | | RCP 6.0 | | | RCP 8.5 | | |
|---|---|---|---|---|---|---|---|---|---|---|---|---|---|---|---|---|
| | $\alpha_{Br}$ | $\eta_{Cl}$ | $\eta_{Br}$ | | $\alpha_{Br}$ | $\eta_{Cl}$ | $\eta_{Br}$ | $\alpha_{Br}$ | $\eta_{Cl}$ | $\eta_{Br}$ | $\alpha_{Br}$ | $\eta_{Cl}$ | $\eta_{Br}$ | $\alpha_{Br}$ | $\eta_{Cl}$ | $\eta_{Br}$ |
| 1980 | 70 | 1.0 | 70 | 2020 | 75 | 0.96 | 72 | 75 | 0.94 | 71 | 74 | 0.93 | 69 | 73 | 0.91 | 67 |
| 1990 | 74 | 0.99 | 74 | 2030 | 75 | 0.94 | 70 | 75 | 0.92 | 69 | 72 | 0.95 | 69 | 72 | 0.93 | 67 |
| 2000 | 76 | 0.97 | 73 | 2040 | 73 | 0.94 | 69 | 73 | 0.90 | 66 | 71 | 0.96 | 68 | 71 | 0.91 | 65 |
| 2010 | 75 | 0.94 | 71 | 2050 | 72 | 0.95 | 68 | 72 | 0.89 | 64 | 70 | 0.93 | 65 | 72 | 0.84 | 60 |
| | | | | 2060 | 70 | 0.96 | 67 | 71 | 0.88 | 63 | 70 | 0.90 | 63 | 72 | 0.78 | 56 |
| | | | | 2070 | 69 | 0.96 | 67 | 70 | 0.87 | 61 | 69 | 0.89 | 61 | 72 | 0.74 | 54 |
| | | | | 2080 | 67 | 0.98 | 66 | 69 | 0.88 | 61 | 69 | 0.85 | 58 | 73 | 0.70 | 51 |
| | | | | 2090 | 66 | 0.98 | 65 | 68 | 0.88 | 60 | 67 | 0.85 | 57 | 73 | 0.67 | 49 |
| | | | | 2100 | 65 | 0.99 | 64 | 66 | 0.87 | 57 | 67 | 0.84 | 57 | 73 | 0.65 | 47 |

[a] $\alpha_{Br}$ calculated per Eq. (4).

[b] $\eta_{Cl}$ calculated per Eq. (6) and $\eta_{Br}$ calculated per Eq. (7), $\Xi = 1980$.

Historical temperature fields obtained from Fleming et al. (1999).

Historical and future greenhouse gas emissions specified per Meinshausen et al. (2011).

Future temperature fields derived from Watanabe et al. (2011).

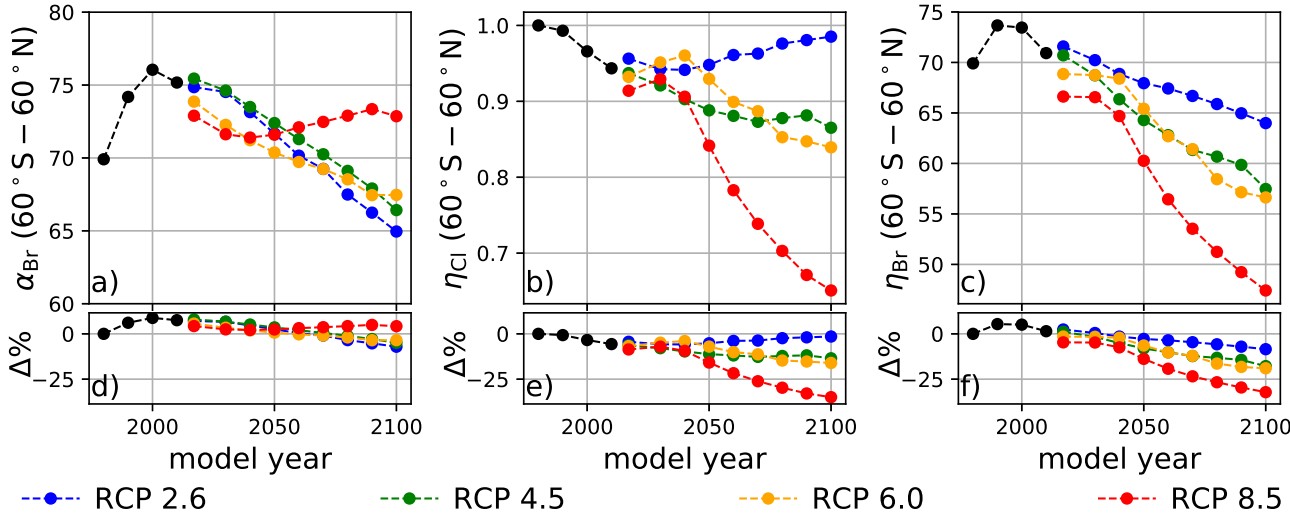

**Figure 3.** Extrapolar $\alpha$ and $\eta$ computed on a decadal basis as a function of RCP scenario. Black traces were calculated using historical boundary conditions. (a) $\alpha_{Br}$, (b) $\eta_{Cl}$, and (c) $\eta_{Br}$. Percent differences of values in (a), (b), and (c) relative to the year 1980 are presented in panels (d), (e), and (f) respectively.

presented in panel (d), that the corresponding $\eta_{Cl}$ and $\eta_{Br}$ values are generally observed to decline by a much more significant extent in panels (e) and (f), respectively.

We highlight the clear qualitative trends of decreasing $\eta_{Cl}$ and $\eta_{Br}$ with climatological forcing scenario severity. Particular notice should be directed to results corresponding to RCP 8.5, in which $\alpha_{Br}$ does not demonstrate a significant coefficient of
variation throughout the 21st century ($CV(\alpha_{Br}) = 2.1\%$), but $\eta$ factors decline precipitously as the century progresses ($CV(\eta_{Cl})$ = 14.3% and $CV(\eta_{Br})$ = 14.5%). This behavior follows the sensitivity expected when there is a large increase in $N_2O$ and $CH_4$. A downward trend in $\alpha_{Br}$ is observed as time propagates in RCP scenarios 2.6, 4.5, and 6.0. This effect is dominated by the declining availability of $ClO_x$ as a result of the slow decay of long-lived ozone-depleting substances. In the case of RCP 2.6, a slight increase in $\eta_{Cl}$ occurs after the year 2040, driven by the continuous decline in the $CH_4$ mixing ratio and the stabilization
of the $N_2O$ mixing ratio in the scenario prescription.

A sensitivity analysis was conducted on $\alpha_{Br}$, $\eta_{Cl}$, and $\eta_{Br}$ in order to clarify the differences presented in Figure 3. In this analysis, $\alpha_{Br}$, $\eta_{Cl}$, and $\eta_{Br}$ were calculated in the manner previously described, but with the chemistry-climate perturbation, $\xi$, identical to the chemoclimatic benchmark, $\Xi$, except for a single perturbed parameter. Four variables were perturbed separately: (a) $N_2O$, (b) $CH_4$, (c) $Br_y/Cl_y$ ratio, and (d) temperature profile. For each perturbation experiment, all factors except for the
perturbed factor were constrained to their year 1980 value(s). Perturbation values were intentionally selected to induce large variations in model response. For $N_2O$ and $CH_4$, mixing ratios were scaled between pre-industrial values and twice the RCP 8.5 year 2100 value. $Br_y/Cl_y$ ratios were selected to range between low values representative of the year 1990, moderate values representative of the year 2000, and high values representative of the year 2100 WMO Table 6-4 projections. Stratospheric




temperature profile perturbations spanned a minimum as parameterized by the RCP 8.5 year 2100 projection to a maximum

value representative of the climatological average of the years 1978-2004.

    Panel (a) of Figure 4 demonstrates that $\alpha_{Br}$ is only slightly sensitive to changes in the mixing ratio of $N_2O$ between pre-industrial and 2x RCP 8.5 year 2100 levels. Unlike $\alpha_{Br}$, both $\eta_{Cl}$ and $\eta_{Br}$, as shown in panels (e) and (i), decline monotonically and with nearly identical gradients, as both the bromine and chlorine cycles are suppressed through reactions with $NO_x$. This suppression arises primarily via the direct formation of the halogen nitrate, as in reaction R17, but also due to a reduction in

the availability of $HO_x$ reaction partners as a result of reaction R18.

$$XO + NO_2 + M \xrightarrow{X=Cl,Br} XONO_2 + M \tag{R17}$$

$$HO + NO_2 + M \rightarrow HONO_2 + M \tag{R18}$$

    Variation in the model output as a function of the mixing ratio of $CH_4$ is presented in Figure 4 panels (b),(f), and (j). Unlike

the case of $N_2O$, $\alpha_{Br}$, panel (b), is a strong function of $CH_4$, increasing as the mixing ratio is increased from the pre-industrial value to 2x RCP 8.5 year 2100 quantities. The reason for this behavior is made evident upon evaluation of $\eta_{Cl}$ and $\eta_{Br}$ in panels (f) and (j). The reaction of Cl with $CH_4$ is fast, forming the inorganic reservoir HCl, but the analogous reaction of Br with $CH_4$ does not effectively occur. Despite these factors, some suppression of the bromine cycle does occur as a result of competition with enhanced $HO_x$ (from the oxidation of $CH_4$) for a reduced quantity of $ClO_x$ reaction partners.

The effect of changing $Br_y/Cl_y$ ratios was investigated over the range of 0.0054 – 0.0088. This range encapsulates the minimum and maximum ratios expected between the years 1980 – 2100 according to WMO 2018 Table 6-4. These values were computed according to Eq. (11) using halocarbon mixing ratios prescribed by WMO 2018 Table 6-4, the fractional release factors of Newman et al. (2006), and an age spectrum of the form of Hall and Plumb (1994). The values of $\alpha_{Br}$, $\eta_{Cl}$, and $\eta_{Br}$ are presented in Figure 4 panels (c), (g), and (k), respectively. Values of $\alpha_{Br}$ generally decrease as the ratio of $Br_y/Cl_y$

increases.

$$\frac{Br_y}{Cl_y}(t,\Gamma) = \frac{\sum_i f_i(\Gamma)\, n_{i,Br} \int_0^\infty \chi_{0,i}(t-t')\, G(t',\Gamma)\, dt'}{\sum_i f_i(\Gamma)\, n_{i,Cl} \int_0^\infty \chi_{0,i}(t-t')\, G(t',\Gamma)\, dt'} \tag{11}$$

    Danilin et al. (1996) demonstrated that $\alpha_{Br}$ in the polar vortex is highly dependent on the relative mixing ratios of available bromine and chlorine, maximizing at low $Br_y/Cl_y$ because of the enhanced abundance of ClO reaction partners for each BrO radical in those conditions. Within the polar vortex the fraction of ozone loss due to the slower chlorine peroxide cycle declines

as $Br_y/Cl_y$ increases; however, the extent of ozone depletion following the addition of bromine does not increase proportionately because the system is controlled by the chlorine abundance. While the same relationship between $\alpha_{Br}$ and $Br_y/Cl_y$ exists in the extrapolar stratosphere, the chemistry responsible for this effect is different. The higher temperatures of the extrapolar



**Figure 4.** Extrapolar $\alpha$ and $\eta$ sensitivity to perturbation parameters. $N_2O$: (a) $\alpha_{Br}$, (e) $\eta_{Cl}$, (i) $\eta_{Br}$. $CH_4$: (b) $\alpha_{Br}$, (f) $\eta_{Cl}$, (j) $\eta_{Br}$. $Br_y/Cl_y$: (c) $\alpha_{Br}$, (g) $\eta_{Cl}$, (k) $\eta_{Br}$. Temperature: (d) $\alpha_{Br}$, (h) $\eta_{Cl}$, (l) $\eta_{Br}$.





stratosphere render the chlorine peroxide cycle ineffective for the loss of ozone. We find that the extent of ozone loss following the addition of bromine increases significantly due to $BrO_x$-$ClO_x$ and $BrO_x$-$HO_x$ cycles rather than staying essentially constant

as in the polar vortex conditions of Danilin et al. (1996).

Evaluations of the model in which stratospheric temperature profiles were varied between RCP 8.5 year 2100 (low), RCP 2.6 year 2030 (medium), and 1978 – 2004 climatological averages (high) demonstrate a dampened sensitivity of $\alpha_{Br}$, as presented in Figure 4 panel (d), in which $\alpha_{Br}$ increases by only 6% from the coldest scenario to the warmest scenario. Heterogeneous activation of chlorine in the coldest scenario boosts $\eta_{Cl}$ by about 17%, as shown in panel (h). The heterogeneous conversion

of bromine reservoirs to active bromine is much less temperature-sensitive than the analogous reactions for chlorine, and this insensitivity is indicated in panel (l); however, $\eta_{Br}$ does respond to the temperature perturbation primarily as a function of changes in the partitioning of $Cl_y$, as in the sensitivity studies of $CH_4$ and $Br_y$/$Cl_y$.

### 3.3   Future EESC

Propagation of EESC using climate-invariant $\alpha_{Br}$ per Eq. (3) or climate-varying $\alpha_{Br}$ per Eq. (5) produces significantly different

dates of extrapolar halogen recovery than propagation of EESC using $\eta$-factor normalization as in Eq. (9). EESC values are presented in Table 3 for historical and future chemistry-climate scenarios. In all cases, EESC compuations were informed by the time-independent fractional release factors provided in Table 1 of Engel et al. (2018). These EESC calculations are visualized in Figure 5.

In panel (a) of Figure 5, EESC is computed per Eq. (3) for static $\alpha_{Br} = 60$ (grey dashed line) and $\alpha_{Br} = 70$ (magenta dashed

line) and Eq. (5) using climate-varying $\alpha_{Br}$ for the four RCP scenarios (colored solid lines). Values of $\alpha_{Br}$ were interpolated between values indicated in Table 2. For reference, the black dots indicate 1980 EESC mixing ratios with $\alpha_{Br}$=70. Notably, there exists very little variation between the RCP scenarios, with maximum deviation of 1.5 years (spanning January 2062 – June 2063) for recovery to 1980 EESC values, as shown in Table 4. Scenarios of climate-invariant $\alpha_{Br}$=60 or $\alpha_{Br}$=70 provide EESC recovery dates (June 2061 and March 2062, respectively) in close agreement with the RCP scenarios. Note that for

clarity, the 1980 EESC reference line for $\alpha_{Br}$=60 is not plotted in Figure 5 panel (a).

Taking chemistry-climate changes into account (when Eq. (9) is used for EESC computation) results in significant variations in future EESC between the RCP scenarios, as shown in panel (b) of Figure 5. For comparison purposes, as in panel (a), the black dots provide the 1980 benchmark EESC mixing ratio with $\alpha_{Br}$=70, and the dashed magenta line shows EESC propagated with climate-invariant $\alpha_{Br} = 70$ (equivalently calculated here with $\eta_{Cl}$ =1 and $\eta_{Br}$ =70). The range of values for the return of

EESC to 1980 levels between the RCP scenarios in panel (b) spans a decade, 2048 to 2058, as shown in Table 4. For all RCP scenarios, the expected recovery date of the inorganic halogen content of the stratosphere to the ozone-depleting equivalent of the year 1980 is significantly sooner than the date expected using $\alpha_{Br}$. Importantly, the earlier ozone recovery dates predicted with our eta factor method using Eq. (9) are in closer agreement with the 3-D CCM results of Dhomse et al. (2018) than EESC recovery dates calculated using bromine alpha factors. We note that this analysis does not include the impact of an accelerated

Brewer-Dobson circulation, which would further hasten our projected date of recovery.




**Table 3.** EESC (pptv) for historical and future chemistry–climate states[a]

| Year | previous method[b] $\alpha_{Br} = 60$ | $\alpha_{Br} = 70$ | Historical[c] $\alpha_{Br}$ | $\eta$ |
|---|---|---|---|---|
| 1980 | 1067 | 1115 | 1115 | 1115 |
| 1990 | 1575 | 1642 | 1664 | 1663 |
| 2000 | 1911 | 2003 | 2056 | 1992 |
| 2010 | 1757 | 1848 | 1896 | 1793 |

| Year | previous method[b] $\alpha_{Br} = 60$ | $\alpha_{Br} = 70$ | RCP 2.6[c] $\alpha_{Br}$ | $\eta$ | RCP 4.5[c] $\alpha_{Br}$ | $\eta$ | RCP 6.0[c] $\alpha_{Br}$ | $\eta$ | RCP 8.5[c] $\alpha_{Br}$ | $\eta$ |
|---|---|---|---|---|---|---|---|---|---|---|
| 2020 | 1614 | 1694 | 1733 | 1653 | 1737 | 1623 | 1723 | 1611 | 1716 | 1572 |
| 2030 | 1478 | 1550 | 1582 | 1493 | 1583 | 1460 | 1567 | 1488 | 1562 | 1450 |
| 2040 | 1335 | 1399 | 1420 | 1337 | 1423 | 1287 | 1408 | 1351 | 1408 | 1279 |
| 2050 | 1198 | 1257 | 1268 | 1201 | 1272 | 1132 | 1260 | 1175 | 1267 | 1074 |
| 2060 | 1084 | 1139 | 1140 | 1094 | 1146 | 1011 | 1137 | 1026 | 1150 | 907 |
| 2070 | 990 | 1042 | 1039 | 1000 | 1049 | 917 | 1039 | 922 | 1055 | 784 |
| 2080 | 914 | 964 | 952 | 928 | 960 | 843 | 957 | 819 | 978 | 691 |
| 2090 | 851 | 899 | 881 | 864 | 889 | 784 | 887 | 752 | 915 | 617 |
| 2100 | 798 | 845 | 822 | 810 | 829 | 719 | 833 | 700 | 859 | 561 |

[a] Stratospheric mean age-of-air = 3 years.

[b] EESC using static $\alpha_{Br}$ calculated per Eq. (3).

[c] EESC using climate-variant $\alpha_{Br}$ calculated per Eq. (5), EESC using $\eta$ calculated per Eq. (9) and benchmarked to $\Xi$=1980.



**Table 4.** Date of EESC recovery to 1980 Benchmark Value[a]

| previous method[b] | | RCP 2.6[c] | | RCP 4.5[c] | | RCP 6.0[c] | | RCP 8.5[c] | |
|---|---|---|---|---|---|---|---|---|---|
| $\alpha_{Br} = 60$ | $\alpha_{Br} = 70$ | $\alpha_{Br}$ | $\eta$ | $\alpha_{Br}$ | $\eta$ | $\alpha_{Br}$ | $\eta$ | $\alpha_{Br}$ | $\eta$ |
| 2061.6 | 2062.2 | 2062.3 | 2057.9 | 2063.0 | 2051.1 | 2062.0 | 2053.7 | 2063.5 | 2047.9 |

[a] Stratospheric mean age-of-air = 3 years. Fractional dates provided to better demonstrate sensitivity of perturbation parameters.
[b] EESC using static $\alpha_{Br}$ calculated per Eq. (3)
[c] EESC using climate-variant $\alpha_{Br}$ calculated per Eq. (5), EESC using $\eta$ calculated per Eq. (9) and benchmarked to $\Xi$=1980.

The divergences of expected EESC values between the calculation techniques are even more pronounced as the century unfolds. Panel (c) of Figure 5 provides the differences between EESC calculated using climate-dependent $\alpha_{Br}$ with Eq. (5) and climate-normalized EESC calculated using Eq. (9). As the century ends, our eta factor method shows that there is a deficit exceeding 300 pptv EESC in the RCP 8.5 scenario relative to a calculation of EESC using the alpha factor method. These

differences are negligible in the RCP 2.6 scenario because the greenhouse gas inventory of the RCP 2.6 year 2100 scenario is very similar to the greenhouse gas inventory of the contemporary stratosphere. Intermediate GHG scenarios lie in between these two extremes.

## 4 Conclusions

The future stratosphere will be very different than the stratosphere of today in terms of trace gas loading, temperature structure,

and radiative-dynamical transport. In this work, we used a 2-D chemical-transport/aerosol model to evaluate how differences in the trace gas loading and the temperature structure of the future atmosphere might influence the relative rates at which inorganic halogen species destroy ozone. These differences can be quite large and are very sensitive to the chemistry-climate boundary conditions imposed.

The most significant perturbations of the stratospheric halogen background in the future are likely to arise from geological

impulses. In this work, we provide the framework for adjusting EESC to accommodate changes in the processing rates of both chlorine and bromine driven by climate and chemistry, such that EESC may be employed to predict ozone loss following such an event. Current formulations of the bromine alpha factor obfuscate the fact that rates of ozone destruction by bromine are changing alongside rates of ozone destruction by chlorine. In some cases, as in RCP 8.5, these rates change in concert, producing a time-invariant $\alpha_{Br}$; however, the actual rates of ozone destruction would have changed significantly, producing

an expected return to 1980 values 14 years earlier than predicted using prior formulations of EESC. For this reason, we have refactored the bromine alpha factor in terms of a climate normalization using new eta factors, which provide an indication of the ozone-processing power of the atmosphere relative to a benchmark date.

Inserting $\eta_{Cl}$ and $\eta_{Br}$ into the formulation for the time-propagation of EESC, as in Eq. (9), teases out differences in the capability of the inorganic halogen background of the stratosphere to destroy ozone as a function of future climate scenario.

Using this treatment, we find that the emission of large quantities of $CH_4$ and $N_2O$, as in the RCP 8.5 emission scenario,





**Figure 5.** Calculations of EESC from 1980 – 2100 using 3-year stratospheric mean age. (a) EESC calculated per Eq. (3) and Eq. (5). Dashed traces: constant $\alpha_{Br}$ as indicated in the legend. Solid traces: $\alpha_{Br}$ interpolated as a function of time from values indicated in Table 2. Dotted black line: EESC corresponding to the year 1980 with $\alpha_{Br}$=70. (b) Calculation of climate-normalized EESC per Eq. (9) with benchmark date $\Xi$ = 1980. Solid lines: $\eta_{Cl}$ and $\eta_{Br}$ interpolated as a function of time from values indicated Table 2. Magenta dashed line: EESC propagated with static $\eta_{Cl}$=1 and $\eta_{Br}$=70 (equivalent to $\alpha_{Br}$=70). Dotted black line: EESC corresponding to the year 1980 with $\alpha_{Br}$=70. (c) RCP scenario differences between panel (a) and panel (b).





decreases the ozone-processing power of the end-of-century future atmosphere by 36% relative to what would be expected by calculating EESC using $\alpha_{Br}$ only (as in Eq. (5)). Our chemistry-climate correction to the current method of calculating EESC brings EESC-parameterized estimates of the extrapolar ozone recovery date into closer agreement with more costly 3-D CCM simulations.

*Author contributions.* Concept: JEK and DMW. Experiment Design: JEK, DKW, RJS, and DMW. Implementation and data reduction: JEK and RJS. All authors discussed the results and developed the manuscript.

*Competing interests.* The authors declare they have no conflict of interest.

*Acknowledgements.* Work conducted at Harvard University was supported by the National Science Foundation under Grant No. 1764171. Work conducted at the University of Maryland was supported by the National Aeronautics and Space Administration Atmospheric Compo-
sition and Modeling Program (ACMAP) under Grant No. 80NSSC19K0983 . JEK thanks N. Allen, J. G. Anderson, W. T. Ball, G. Chiodo, J. Hansen, and T. Peter for helpful discussion of the work. For our use of RCP temperature fields, we acknowledge the World Climate Research Programme's Working Group on Coupled Modelling, which is responsible for CMIP, and we thank the Japan Agency for MarineEarth Science and Technology, Atmosphere and Ocean Research Institute (The University of Tokyo), and National Institute for Environmental Studies.





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
