# Peer review of "Reformulating the Bromine Alpha Factor and EESC: Evolution of Ozone Destruction Rates of Bromine and Chlorine in Future Climate Scenarios"

_Atmospheric Chemistry and Physics, 2020_

## Referee Comment (RC1) · Andreas Engel (Referee) · 30 Apr 2020

This paper presents an interesting study on the change of the relative efficiency of Bromine with respect to Chlorine Atoms in destroying ozone under varying atmospheric conditions and of the changing impact of halogens on ozone in future climates. The title has its emphasis on Bromine, but actually the changing efficiency of Chlorine is also studies, so the authors might consider adjusting the title to take this into account. The subject is well in the scope of ACP and worth studying. However, I believe that the authors are extending the EESC concept to something it is not: a proxy for the effect of halogens on ozone even under changing climate and chemical conditions and for

projection of ozone recovery. It is discussed in the last WMO assessment that EESC is a proxy for the inorganic halogen content and its recovery should not be considered to be a recovery of the ozone layer. Therefore, while I find the new concept useful, I suggest to not mix it with the concept of EESC. In addition, an extremely important further driver (namely the change in Brewer-Dobson circulation) is not included in the concept. Therefore, this new concept extends EESC into the direction of an ozone proxy, yet it only takes into account a part of the expected climate-related changes, namely those arising from different temperatures and those from different chemical regimes. Dynamical changes (which are highly uncertain but may be very large drivers) are not incorporated, which makes this new concept difficult to position: it is neither a halogen proxy under otherwise unchanged conditions (which EESC is), nor is it a real proxy for ozone recovery (as it lacks the dynamical changes). I also have some more specific comments, which are listed below. I can recommend the paper for publication after revisions as to my specific comments and also after a more thorough discussion what this proxy really represents and after considering whether this new proxy should still be called EESC (I believe it should not be called EESC).

Specific comments

Major comments

(i) As explained above, I believe that the authors should give some more consideration to what the new proxy really is and if it should still be named EESC, in particular as not only the $\alpha$ value for bromine becomes a variably, but also the effectiveness of chlorine to destroy ozone ($\eta$) becomes a variable. This was not considered in EESC so far as EESC so far was not really an ozone recovery proxy (see e.g. the discussion in box 1-4 of WMO 2018), even though it has often been used as such. I would consider naming this differently, maybe something like "Equivalent Ozone-effective stratospheric chlorine (EOESC)" or something along this line. Further, the nice thing about EESC as it is used now is that it can be easily calculated and applied to a new scenario, without the need for a model. This advantage is lost with the new concept, unless a method to

parametrise the $\eta$ factors needed in eq. (9) is given.

(ii) The study is performed with a 2D model. How is the climate state taken into account, i.e. the change in temperature and the changes in dynamics due to the expected accelerated BDC? While the changes in temperature and chemical environment can be simulated in a 2D model, the change in the Brewer-Dobson circulation which is projected by the 3D climate models is most probably not included. Under such changing climatic conditions, the new fractional release factor formulation by Ostermöller et al., (2017) is independent of the trend of the species but it does depend on the state of the atmosphere, in particular it is expected to change with time for a given mean age value due to dynamical changes (accelerating BDC).

(iii) Unfortunately, the explanation of the model experiments is rather unclear and difficult to follow. The paper lacks a clear explanation of which model runs have been performed, and exactly how they have been forced. In particular, the dynamical forcing is not described and it is unclear if changes in the Brewer-Dobson circulation are included in the simulations from the description in section 2. Only on p.15, l.329 it is clearly stated that changes in the BDC are not included. A clearer description is required here to ensure that the results can be understood and reproduced.

(iv) In section 3.1. it would be important to describe more clearly the physical meaning of the new EESC formulation (9). The definition of the $\eta$ values for chlorine and bromine is always relative to the Ozone sensitivity with respect to Cl in the reference state. Therefore, EESC defined in (9) is also referenced to dO3/dCl in that reference state. It would be good to explain this concept more clearly and give a more intuitive explanation of this quantity. In my understanding the new formulation in eq. 9 describes the 1980-equivalent stratospheric chlorine impact on ozone, adjusted for changing stratospheric temperature and changing chemical environment, but not for changing dynamics.

Minor comments:

General: the term background is used in many places (e.g. l. 148: inorganic halogen background). A background is a state against which something is referenced. I suppose level or content might be more appropriate.

l. 15.: what is meant by inorganic halogen precursor compounds? I suppose this is the source gases? Then I would term this the precursor compounds for inorganic halogen.

l. 45: the use of "unlike" is unclear to me: in the absence of chlorine, also Br would require the oxygen atom and there are also other Cl-recycling reactions.

l. 49: please specify what you mean by lower stratosphere here.

l. 54 (and other places): please be more specific with respect to the WMO 2018 citations: Usually the respective chapter should be cited in order to allow the reader to find the information.

l. 86: what do you mean by chemoclimatic?

l. 107: see for example discussion in box 1-4 of WMO 2018: EESC should really not be used as an ozone recovery proxy. It is a halogen recovery proxy. See also major comment above.

l. 114-125: the projected super recovery of stratospheric ozone is mainly due to changes in dynamics, not changes in chemistry. This section reads like the chemical influences are dominating.

Section 2: In this section a clearer discussion of the model set-up is required, in particular how the dynamics (and possibly changing dynamics) have been incorporated.

l. 171ff: The concept of the perturbation experiments should be clearer explained.

l. 202-204: A clearer description should be given specifying that both sensitivities are given relative to the sensitivity of ozone to chlorine in the benchmark chemical-climate state.

Section 3.2.: have perturbations in T and in chemistry been performed independently?

i.e. can it be distinguished between an effect due to increased CH4 and increased HOx with respect to an effect due to increased T?

l. 235: I suggest using the term temporal evolution or temporal development instead of trajectory, as trajectory has a different meaning in atmospheric sciences.

l. 243: please give the percentage increases relative to what? Also monotonic and percentual do not go very well together. I suppose what is meant is a linear trend resulting in an increase of xx eq. (11): which time series is used here? If I understand correctly, the model is run for 20 years into equilibrium. In this case, the temporal trend of the trace gas in the integral would be equal to the (constant) mixing ratio and the whole integral would become the (constant) mixing ratio.

l. 312: The values in Table 1 in Engel et al. (2018) are trend-independent. Fractional release factors are expected to change for different climate states.

l. 314 and Figure 5: the grey used here looked very "blue-gray" on my printout. I suggest to use a clearer grey colour for better distinction.

l. 317.: why does the EESC formulation according to Engel et al. show differences for different RCP scenarios at all? Should EESC not be independent of RCP in this formulation?

l. 329.: This information should come much earlier and be discussed in section 2.

l. 344: Can the dominance by geological perturbations (I supposes volcanoes) be substantiated by a reference?

l.345: processing rates of what? I suppose ozone?
* * *

---

## Referee Comment (RC2) · Anonymous Referee #2 · 6 May 2020

Authors: J. Eric Klobas, Debra K. Weisenstein, Ross J. Salawitch, and David M. Wilmouth.

Title: Temporal Evolution of the Bromine Alpha Factor and Equivalent Effective Stratospheric Chlorine in Future Climate Scenarios.

General Comment: This is a very interesting paper that attempts to modify the Equivalent Effective Stratospheric Chlorine (EESC) definition to include the effect of possible future climate scenarios. This is done by "refactoring" the bromine alpha factor introducing a "climate normalization to a benchmark climate state." This is changing the simple purpose (see WMO definition below) of the EESC to represent a much broader

definition of climate influence on ODS levels in the stratosphere.

As defined by the WMO 2018 assessment executive summary: "EESC is a metric for representing ODS levels in the stratosphere. It is calculated based upon three factors: surface atmospheric concentrations of individual ODSs and their number of chlorine and bromine atoms, the relative efficiency of chlorine and bromine for ozone depletion, and the time required for the substances to reach different stratospheric regions and break down to release their chlorine and bromine atoms. As EESC continues to decrease in response to Montreal Protocol provisions, stratospheric ozone is expected to increase. In this Assessment, EESC does not include chlorine and bromine from very short-lived substances (VSLSs)."

My feeling is that one should not change the EESC definition. However, that being said, I do think there is merit in attempting to include a diagnostic that does address climate impacts on EESC, that is simple, and does not require running a large ensemble of CCMs. Therefore, I wouldn't change the EESC definition above, but would create a new definition. This work is a first step towards this goal. I would recommend that this work be published assuming my comments are addressed below.

Specific Comments:

Line 35: I don't believe (just a suggestion) you need to discuss homogeneous reactions (i.e., like R4) in discussion of lower stratospheric ozone loss. This is a topic that has been explained in hundreds of publications. Just reference the Solomon et al., 1999 review article. You also don't need to summarize the heterogeneous reactions either (i.e., R5-R7).

Lines 135-144: RE: Discussion of Volcanic emission of Cl and Br. I find this discussion topic distracts from the point of this paper. Why go into possible random inputs of these species into a future atmosphere. You might as well discuss the possibility of an ocean surface asteroid impact injecting Cl and Br into the stratosphere. This topic seems like a separate study/discussion to me. I would just focus on the modified "EESC"

[Figure]

technique you are proposing.

Lines 152-163: The model description section is very confusing (at first read). One has to have a basic understanding of Daniel et al., 1999 to make sense on where you are going with the scenarios. Evidently you are running time slice experiments (every 10-years, with a duration of 20 years) using constant mole fraction lower boundary conditions for the 20-year period? E.g., Table 1: for "d" superscript you state "informed by Meinshausen et al. (2011) and Watanabe et al. (2011)". This means you are getting the initial conditions for say year 2020 from Watenabe et al. and the lower boundary mole fraction from Meinshausen et al.? For "c" you are not using the same model, but a 2D model from Fleming et al., (1999)? Why not use the same model for hindcast and future conditions (i.e., MIROC-CHEM-ESM)?

Line 164: You state that you are using the Daniel et al. (1999) approach. Essentially you are using the approach for equation (2) in Daniel et al., correct? [Your equation (2)] This is also why you have three scenarios to derive alpha-Br from a given atmospheric state, correct? I would restate (in your words) the procedure on page 23,874 Daniel et al. (1999). This will greatly help the first-time reader of this work.

Line 218: You probably should define the basic technique of graph-theory.

Line 229: Specified dynamics details are needed here. What are you specifying for the dynamical fields and where did they come from?

Lines 325-330. This is a very interesting result [i.e., better comparison of EESC to 1980 values compared to Dhomse et al. (2018)]. The Dhomse et al. study was an average of many models. Have you looked at one model, say the MIROC-CHEM-ESM, of which was used for the initial condition, for this work?

Lines 329-330. You state that this analysis does not include the "impact of an accelerated BDC, which would hasten the projected recovery". Since you are using a CCM for your initial state, is part of this process "baked into" the calculation? Certainly, the

temperature affect is; but isn't it possible that the dynamical state is also influencing the equation 10 result?

NOTE: I would find it very interesting to add an additional figure (like Figure 1) showing the column alpha-Br (latitude vs time) for year 2100. Here I would show four panels, depicting the result for RCP2.6, 4.5, 6.0, and 8.5.
* * *

---

## Author Comment (AC1) · 5 Jun 2020

We thank the referee for his thorough and thoughtful remarks. We have revised our manuscript accordingly. The referee's comments are presented below in **bold text** and our responses to the referee appear in plain text.

**The title has its emphasis on Bromine, but actually the changing efficiency of Chlorine is also studies, so the authors might consider adjusting the title to take this into account.**

Good suggestion. We have changed the title to:

[Figure]

Reformulating the Bromine Alpha Factor and EESC: Evolution of Ozone Destruction Rates of Bromine and Chlorine in Future Climate Scenarios

**(i) As explained above, I believe that the authors should give some more consideration to what the new proxy really is and if it should still be named EESC, in particular as not only the $\alpha$ value for bromine becomes a variably, but also the effectiveness of chlorine to destroy ozone ($\eta$) becomes a variable. This was not considered in EESC so far as EESC so far was not really an ozone recovery proxy (see e.g. the discussion in box 1-4 of WMO 2018), even though it has often been used as such.**

**I would consider naming this differently, maybe something like "Equivalent Ozone-effective stratospheric chlorine (EOESC)" or something along this line.**

We agree with the referee that Equation (9) provides a quantity that can be differentiated from prior definitions of EESC. Equation (9) is a scalar multiplication of EESC with the chlorine eta factor (effectively a benchmark-state normalization of chlorine). We now call this metric Equivalent Effective Stratospheric Benchmark-normalized Chlorine (EESBnC).

**Further, the nice thing about EESC as it is used now is that it can be easily calculated and applied to a new scenario, without the need for a model. This advantage is lost with the new concept, unless a method to parametrise the $\eta$ factors needed in eq. (9) is given.**

We note that our method adds no further complication to the calculation of EESC (or our new proxy) than is already present. The alpha factor is itself a parameterized quantity derived from 2-D modeling studies.

**(ii) The study is performed with a 2D model.**

Nearly all prior model determinations of alpha factor that we are aware of were also computed with 2-D models, e.g., Danilin et al., 1996, Ko et al., 1998, Daniel et al.,

1999, Sinnhuber et al., 2009, and unpublished results discussed in Chapter 8 of the 2006 WMO Ozone Assessment. We decided to employ the AER-2D model for this work because it (a) provides a direct linkage with prior determinations of the bromine alpha factor for validation purposes, (b) provides adequate spatial and temporal resolution for the determination of regional-annual parameterizations of the alpha- and eta-factors, and (c) provides these results with reasonable computational cost scaling. We note that the results presented in this work constitute over 2160 model years of evaluation, requiring more than a year-and-a-half of single-threaded computing time on the Harvard Cannon supercomputer. Because we report regional-annual average phenomena which are reproduced quite well by the model (Weisenstein et al., 1997, Weisenstein et al., 2007), the quality of our results are not materially degraded relative to the results we would have obtained if we had employed a 3-D model at significantly higher computational cost.

**How is the climate state taken into account, i.e. the change in temperature and the changes in dynamics due to the expected accelerated BDC? While the changes in temperature and chemical environment can be simulated in a 2D model, the change in the Brewer-Dobson circulation which is projected by the 3D climate models is most probably not included.**

Correct, we do not include changes in the Brewer-Dobson circulation. Our methodology, as outlined on lines 229 - 234 of the original manuscript, employs specified dynamics based upon a climatology from 1978 to 2004. The inclusion of varying circulation patterns is interesting; such a study is complicated because non-local factors influencing ozone mixing ratios might no longer be negligible when comparing the climate perturbation and climate benchmark scenario, and the magnitude of these non-local effects will likely differ between various models. A quantification of the effect of an accelerating Brewer-Dobson circulation on alpha- and eta-factors would be valuable and interesting on its own and in relation to this work.

**Under such changing climatic conditions, the new fractional release factor for-**

**mulation by Ostermöller et al., (2017) is independent of the trend of the species but it does depend on the state of the atmosphere, in particular it is expected to change with time for a given mean age value due to dynamical changes (accelerating BDC).**

We acknowledge the dependence of the fractional release factor on the synoptic circulation and have added text to discuss this on lines 270 - 273 of the revised manuscript:

In all cases, the computations were informed by the trend-independent fractional release factors provided in Table 1 of Engel et al. (2018a); though fractional release factors are likely to vary as the climate evolves (Leedham-Elvidge et al., 2018), these factors correlate with the specified dynamics employed in this analysis.

We note that our work uses specified dynamics corresponding to the circulation patterns of the contemporary era. In the context of our analysis, which provides ozone-loss processing rates as a function of changing temperature and trace-gas inventories, but not changing circulation, the fractional release factors would vary in only a slight manner, due to small changes in certain chemical terms such at the rate of halocarbon activation expected following stratospheric cooling [for example: CFC-12 + O1D -> ClO + products: k(240 K) =1.55E-10, k(230 K) = 1.56E-10 using JPL-2015 kinetics]. Consequently, the fractional release factors provided by Ostermöller et al. (2017) are appropriate fractional release factors to employ in our analysis.

**(iii) Unfortunately, the explanation of the model experiments is rather unclear and difficult to follow. The paper lacks a clear explanation of which model runs have been performed, and exactly how they have been forced.**

We have added further text to the experiment description beginning on line 164 of the original manuscript in an effort to more clearly describe the experiments that were performed. We note that our discussion on the treatment of data to derive the reported quantities of the bromine alpha factor and the chlorine and bromine eta factors is described elsewhere in the text. This is because our definition of alpha factor and eta

factor are novel and must be introduced first.

**In particular, the dynamical forcing is not described and it is unclear if changes in the Brewer-Dobson circulation are included in the simulations from the description in section 2. Only on p.15, l.329 it is clearly stated that changes in the BDC are not included. A clearer description is required here to ensure that the results can be understood and reproduced.**

We have revised the text to explicitly indicate that specified dynamics were employed in the model description. The perturbation experiments are now described more explicitly.

**(iv) In section 3.1. it would be important to describe more clearly the physical meaning of the new EESC formulation (9). The definition of the $\eta$ values for chlorine and bromine is always relative to the Ozone sensitivity with respect to Cl in the reference state. Therefore, EESC defined in (9) is also referenced to dO3/dCl in that reference state. It would be good to explain this concept more clearly and give a more intuitive explanation of this quantity. In my understanding the new formulation in eq. 9 describes the 1980- equivalent stratospheric chlorine impact on ozone, adjusted for changing stratospheric temperature and changing chemical environment, but not for changing dynamics.**

We have added a more intuitive discussion of the meaning of Eq. (9) to section 3.1 on lines 176 - 180 of the revised manuscript:

The difference between EESC and EESBnC is significant; whereas EESC considers "the *relative* efficiency of chlorine and bromine for ozone depletion" (World Meteorological Organization, 2018), EESBnC accounts for the *overall* efficiency of chlorine and bromine relative to a benchmark chemistry/climate state. Thus, EESBnC provides the ozone-depleting power of an air parcel in the stratosphere propagated independent of changes in the rates of chlorine or bromine ozone-loss catalysis

**Minor comments: General: the term background is used in many places (e.g.**

**l. 148: inorganic halogen background). A background is a state against which something is referenced. I suppose level or content might be more appropriate.**

We have reviewed every instance of the word 'background' and have clarified our meaning when appropriate.

**l. 15.: what is meant by inorganic halogen precursor compounds? I suppose this is the source gases? Then I would term this the precursor compounds for inorganic halogen.**

We have no preference for either phrasing and have adopted the phrasing suggested by the referee.

**l. 45: the use of "unlike" is unclear to me: in the absence of chlorine, also Br would require the oxygen atom and there are also other Cl-recycling reactions.**

Yes, it is useful to consider the chain effectiveness (e.g., Lary (1997)) when comparing catalytic cycles involving chain centers with large differences in mixing ratio. We have revised our discussion to express this concept and relevant citations.

**l. 49: please specify what you mean by lower stratosphere here.**

We would refer the reader to the individual cited documents for the boundaries of the lower stratosphere as they vary between publications.

**l. 54 (and other places): please be more specific with respect to the WMO 2018 citations: Usually the respective chapter should be cited in order to allow the reader to find the Information.**

We have reviewed every instance in which the WMO Scientific Assessment of Ozone Depletion is cited, and now cite individual chapters when the information being cited is found primarily in one chapter. We retain citation to the document as a whole for information which can be found throughout the entire document.

Additionally, we have removed several citations to the WMO Scientific Assessment

of Ozone Depletion because the information cited is trivial: line 15 of the original manuscript, line 54 of the original manuscript.

**l. 86: what do you mean by chemoclimatic?**

Chemoclimatic: of or relating to the confluence of chemical and climatic properties. We have replaced all instances of this word with chemistry/climate.

**l. 107: see for example discussion in box 1-4 of WMO 2018: EESC should really not be used as an ozone recovery proxy. It is a halogen recovery proxy. See also major comment above.**

We agree with the referee that EESC should not be considered an ozone recovery proxy and note that we state that EESC is a halogen recovery proxy ourselves on the same line. That said, EESC is commonly employed to predict the date of, or set limits on the date of, ozone recovery. One key result of this work is to provide a new quantity which is better suited for this purpose.

We thank the referee for suggesting that we differentiate the name of this proxy from EESC.

**l. 114-125: the projected super recovery of stratospheric ozone is mainly due to changes in dynamics, not changes in chemistry. This section reads like the chemical influences are dominating.**

We have modified our discussion to better express that the expected super-recovery is dependent on both photochemical and dynamical controls in different parts of the stratosphere.

**Section 2: In this section a clearer discussion of the model set-up is required, in particular how the dynamics (and possibly changing dynamics) have been incorporated.**

We have revised our model description to provide a clearer understanding of how the

experiment was conducted and how dynamics were incorporated.

**l. 171ff: The concept of the perturbation experiments should be clearer explained.**

We have rewritten portions of this section to more clearly explain the perturbation procedure.

**l. 202-204: A clearer description should be given specifying that both sensitivities are given relative to the sensitivity of ozone to chlorine in the benchmark chemical-climate state.**

We have now clarified the meaning of this this new variable on lines 166-168 of the revised manuscript:

The eta factor thus expresses the ozone-depleting efficiency of a chlorine or bromine atom in an arbitrary chemistry/climate state relative to the ozone-depleting efficiency of a chlorine atom in the benchmark chemistry/climate state.

**Section 3.2.: have perturbations in T and in chemistry been performed independently? i.e. can it be distinguished between an effect due to increased CH$_4$ and increased HO$_x$ with respect to an effect due to increased T?**

Yes, we direct the referee to the sensitivity studies described in the text beginning on line 261 of the original manuscript and summarized in figure 4, where the sensitivity parameters of CH$_4$, N$_2$O, T, and Br$_y$:Cl$_y$ were perturbed independently.

**l. 235: I suggest using the term temporal evolution or temporal development instead of trajectory, as trajectory has a different meaning in atmospheric sciences.**

We do not have a preference for the terminology and have adopted the phrase 'temporal dependencies' per the referee's suggestion.

**l. 243: please give the percentage increases relative to what? Also monotonic**

**and percentual do not go very well together. I suppose what is meant is a linear trend resulting in an increase of xx**

We clarified that the percentage increases are relative to year 1980 and also removed the word monotonically.

**eq. (11): which time series is used here? If I understand correctly, the model is run for 20 years to the (constant) mixing ratio and the whole integral would become the (constant) mixing ratio.**

We thank the referee for bringing this question to our attention. The $\mathrm{Br}_y : Cl_y$ ratios should be computed using a constant tropospheric mixing ratio for each halocarbon species. The quantities have been recalculated and Figure 4 has been regenerated. Eq. (11) has been modified to reflect this. We note that the results of this sensitivity study are not qualitatively changed.

**l. 312: The values in Table 1 in Engel et al. (2018) are trend-independent. Fractional release factors are expected to change for different climate states.**

We have reworded the sentence to express that the FRF are trend-independent and subject to change with future climate evolution.

**l. 314 and Figure 5: the grey used here looked very "blue-gray" on my printout. I suggest to use a clearer grey colour for better distinction**

Figure 5 has been regenerated to change the label on the y-axis of panel (b). We have changed the color of the grey to be less blue in the process.

**l. 317.: why does the EESC formulation according to Engel et al. show differences for different RCP scenarios at all? Should EESC not be independent of RCP in this formulation?**

The EESC formulation according to Engel et al. (eq. (3)) does not take climate as an input parameter. The EESC formulations according to our eq. (5) do take climate as

input parameters and vary according to the RCP scenario. We have modified the sentence to more clearly specify that the plots with climate dependence were calculated according to eq. (5).

**l. 329.: This information should come much earlier and be discussed in section 2.**

Though we state that we employ specified dynamics in several locations in the original text, we now more explicitly discuss this in section 2.

**l. 344: Can the dominance by geological perturbations (I supposes volcanoes) be substantiated by a reference?**

Please refer to Klobas, et al., (2017) and references cited therein for more information regarding the potential of future halogen-rich eruptions to perturb ozone.

These statements have been removed per the anonymous referee's suggestion.

**l.345: processing rates of what? I suppose ozone?**

Yes, we now specify that these rates are ozone-processing rates.

---

## Author Comment (AC2) · 5 Jun 2020

We thank the anonymous referee for his or her thoughtful criticism which has resulted in changes that have improved the quality of our manuscript. We provide point-by-point responses to the referee's comments (**bold text**) in plain text below.

**My feeling is that one should not change the EESC definition. However, that being said, I do think there is merit in attempting to include a diagnostic that does address climate impacts on EESC, that is simple, and does not require running a large ensemble of CCMs. Therefore, I wouldn't change the EESC definition above, but would create a new definition. This work is a first step towards this**

**goal. I would recommend that this work be published assuming my comments are addressed below.**

Per the recommendation of both referees regarding changing the variable name, we have opted to call this proxy Equivalent Effective Stratospheric Benchmark-normalized Chlorine (EESBnC) and will refer to it as such in our responses here.

**Specific Comments: Line 35: I don't believe (just a suggestion) you need to discuss homogeneous reactions (i.e., like R4) in discussion of lower stratospheric ozone loss. This is a topic that has been explained in hundreds of publications. Just reference the Solomon et al., 1999 review article. You also don't need to summarize the heterogeneous reactions either (i.e., R5-R7).**

We have revised the introduction to eliminate them.

**Lines 135-144: RE: Discussion of Volcanic emission of Cl and Br. I find this discussion topic distracts from the point of this paper. Why go into possible random inputs of these species into a future atmosphere. You might as well discuss the possibility of an ocean surface asteroid impact injecting Cl and Br into the stratosphere. This topic seems like a separate study/discussion to me. I would just focus on the modified "EESC" technique you are proposing.**

We have removed this paragraph and the associated statement in the conclusion.

**Lines 152-163: The model description section is very confusing (at first read). One has to have a basic understanding of Daniel et al., 1999 to make sense on where you are going with the scenarios. Evidently you are running time slice experiments (every 10-years, with a duration of 20 years) using constant mole fraction lower boundary conditions for the 20-year period? E.g., Table 1: for "d" superscript you state "informed by Meinshausen et al. (2011) and Watanabe et al. (2011)". This means you are getting the initial conditions for say year 2020 from Watenabe et al. and the lower boundary mole fraction from Meinshausen et**

**al.?**

We have revised the model and experiment description sections to better communicate the procedure we followed. The referee's interpretation is correct. We employ 20-year time slice experiments for each decade spanning 1980 - 2100. Temperature fields are obtained from RCP scenario realizations of the MIROC-CHEM-ESM from the CMIP5 archive (Watanabe et al., 2011), while chemical boundary conditions for our model evaluation are obtained from RCP scenario specifications (Meinshausen et al., 2011).

**For "c" you are not using the same model, but a 2D model from Fleming et al., (1999)? Why not use the same model for hindcast and future conditions (i.e., MIROC-CHEM-ESM)?**

For 'c', historical past simulations were evaluated using climatological and temperature fields prepared previously. We employ these climatological conditions because (1) these climatological fields were used in previous studies and facilitated validation of model performance, and (2) the climatological fields prepared from MIROC-CHEM-ESM for the present do not significantly differ with the climatological fields we employ.

**Line 164: You state that you are using the Daniel et al. (1999) approach. Essentially you are using the approach for equation (2) in Daniel et al., correct? [Your equation (2)] This is also why you have three scenarios to derive alpha-Br from a given atmospheric state, correct? I would restate (in your words) the procedure on page 23,874 Daniel et al. (1999). This will greatly help the first-time reader of this work.**

The reviewer is correct on all counts. We have reworded the procedure such that it is more clearly communicated to a reader who does not have prior knowledge of Daniel et al. (1999).

**Line 218: You probably should define the basic technique of graph-theory.**

We note that the technique is described immediately following this line.

**Line 229: Specified dynamics details are needed here. What are you specifying for the dynamical fields and where did they come from?**

We now provide the information the referee requests in Section 2, model description.

**Lines 325-330. This is a very interesting result [i.e., better comparison of EESC to 1980 values compared to Dhomse et al. (2018)]. The Dhomse et al. study was an average of many models. Have you looked at one model, say the MIROC-CHEM-ESM, of which was used for the initial condition, for this work?**

In response to the referee's question, we performed a quick investigation and found that MIROC-CHEM-ESM RCP / HISTORICAL experiments from CMIP5 show a similar qualitative trend to our results (extrapolar RCP8.5 recovering to 1980 ozone layer thickness sooner and extrapolar RCP 2.6 recovering to 1980 ozone layer thickness later). The dates of recovery from these experiments are not exactly the same as the dates we derive from our EESBnC treatment in Table 4, which is not unexpected given that our prognostication is based upon halocarbon inventories and our eta parameters are derived using a model with different chemical/aerosol/transport schema.

We note that Eyring et al., 2013 explored ozone layer recovery to 1980 thickness and the authors included MIROC-CHEM-ESM in their model ensemble.

**Lines 329-330. You state that this analysis does not include the "impact of an accelerated BDC, which would hasten the projected recovery". Since you are using a CCM for your initial state, is part of this process "baked into" the calculation? Certainly, the temperature affect is; but isn't it possible that the dynamical state is also influencing the equation 10 result?**

It's true that certain parameters such as scale height will be dependent on the imposed temperature structure, such that the dynamics deviate slightly between model realizations. If we review vertical profiles of ozone for the control runs in our temperature sensitivity studies, we find very little variation between midlatitude ozone profiles in the

lower stratosphere where ozone is subject to dynamical control, especially with regard to circulation-induced ozone super-recovery. However, regions where photochemistry dominates the ozone steady-state do exhibit variation in ozone as a function of temperature boundary conditions.

From these comparisons, we infer that any variation in ozone due to dynamics baked-in to our boundary conditions are insignificant to photochemical changes in ozone as a result of the temperature perturbation.

**NOTE: I would find it very interesting to add an additional figure (like Figure 1) showing the column alpha-Br (latitude vs time) for year 2100. Here I would show four panels, depicting the result for RCP2.6, 4.5, 6.0, and 8.5.**

We agree that such a plot is quite interesting, but we are reserving this type of analysis for a future manuscript involving a method which is more sensitive to PSC response to climate changes in the polar regions.